# Recent Advances in Vertically Aligned Nanocomposites with Tunable Optical Anisotropy: Fundamentals and Beyond

**Xuejing Wang [1] and Haiyan Wang [1,2,*]**

1    School of Materials Engineering, Purdue University, West Lafayette, IN 47907, USA; xuejing@lanl.gov
2    School of Electrical and Computer Engineering, Purdue University, West Lafayette, IN 47907, USA
*    Correspondence: hwang00@purdue.edu

**Abstract:** Developing reliable and tunable metamaterials is fundamental to next-generation optical-based nanodevices and computing schemes. In this review, an overview of recent progress made with a unique group of ceramic-based functional nanocomposites, i.e., vertically aligned nanocomposites (VANs), is presented, with the focus on the tunable anisotropic optical properties. Using a self-assembling bottom-up deposition method, the as-grown VANs present great promise in terms of structural flexibility and property tunability. Such broad tunability of functionalities is achieved through VAN designs, material selection, growth control, and strain coupling. The as-grown multi-phase VAN films also present enormous advantages, including wafer scale integration, epitaxial quality, sharp atomic interface, as well as designable materials and geometries. This review also covers the research directions with practical device potentials, such as multiplex sensing, high-temperature plasmonics, magneto-optical switching, as well as photonic circuits.

**Keywords:** vertically aligned nanocomposite (VAN); metamaterial; tunability; pulsed laser deposition (PLD); optical anisotropy

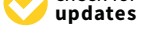



## 1. Optical Anisotropy

Optical anisotropy can be defined as the differences of complex refractive indices along the principal axes when light propagates through media [1]. Photonic crystals possessing birefringence or dichroism phenomena are considered as natural anisotropic media, but are either too bulky or present with limited birefringence [2–4]. Research continue developing structures with broken symmetry, such as layered two-dimensional crystals or transition metal dichalcogenides (TMDs) [5–7], as well as artificially constructed nanostructures or the so-called metamaterials that surpass the conventional limit in achieving extreme anisotropy at controlled frequencies [8–10].

Hyperbolic metamaterial (HMM) is one representative anisotropic medium that displays hyperboloid topologies (Figure 1b,c) at *k*-space, as opposed to the normal isotropic media (Figure 1a) [11–13]. Such anisotropy is normally achieved by coupling dielectric and metallic components at nanoscale as wire-in-matrix or multilayered schemes, which find applications in sensing, spontaneous emission, negative refraction, or designs of superlenses [14–16]. The multilayered HMMs can be further patterned into double-fishnet nanostructures (Figure 1c) for tunable negative refraction, or nanoresonator array (Figure 1d) that is completely fabricated with semiconductor candidates [17–21]. Wire metamaterial, on the other hand, provides additional flexibility, such as density or distribution (Figure 1f), aspect ratio, as well as geometry of the nanowires [8,22–25]. Contrary to the metallic nanowire in dielectric matrix configuration, the plasmonic nanohole structure (Figure 1g) can be considered as periodic "air" (*n* = 1) holes being embedded in the plasmonic media, which has been intensively explored for microfluidic sensing and superlenses [26–29]. Overall, the concept of building artificial nanostructures has already been implemented in multifunctional metamaterials toward sensing, ultrafast switching, spintronics, and

neuromorphic computing [30–33]. For example, plasmonics sensors built by active metamaterials or metasurfaces have been demonstrated with controllable detection frequencies and detection limits that are crucial to biomedical and nanophotonic devices [34,35].

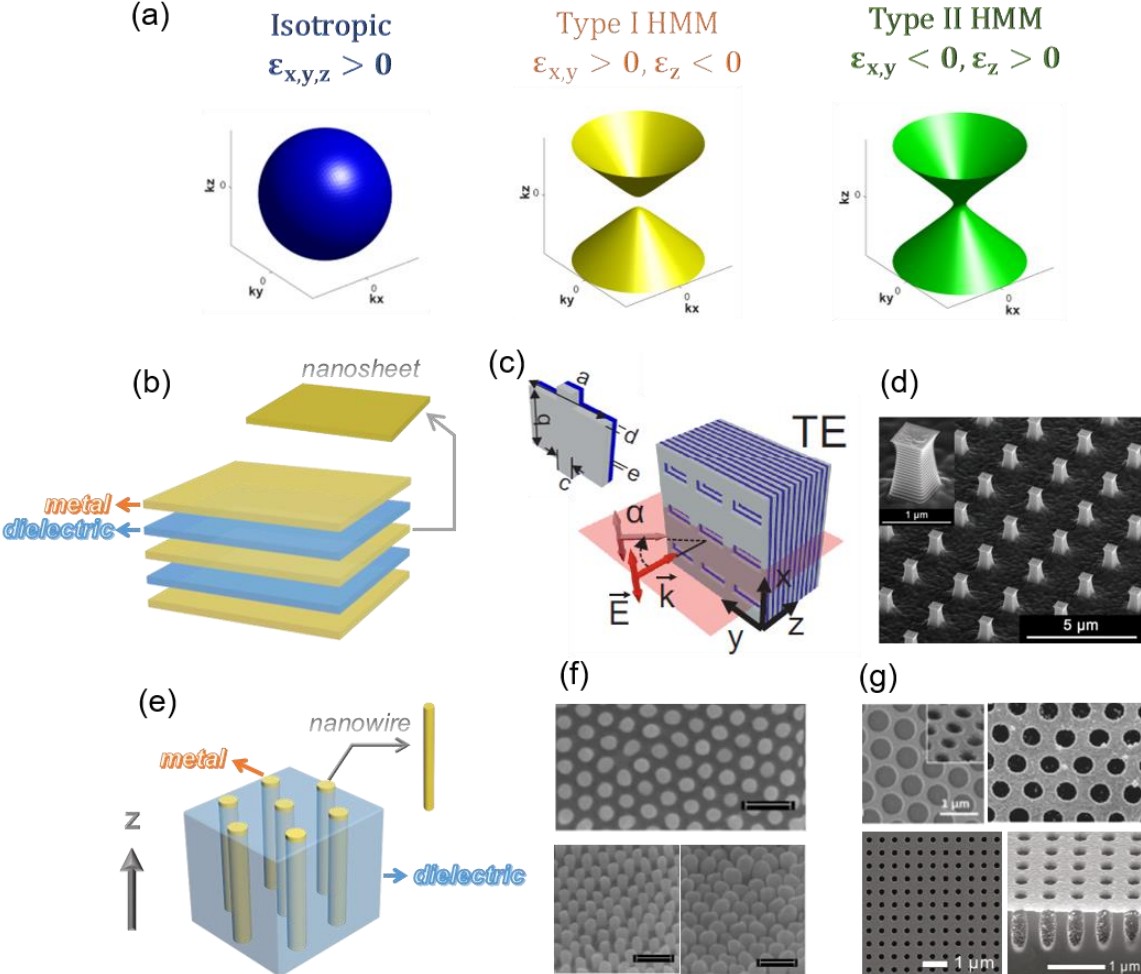

**Figure 1.** Optical metamaterials realizing extreme anisotropy. (**a**) *k*-space topologies of isotropic ($\varepsilon_{x,y,z} > 0$) materials, Type I ($\varepsilon_{x,y} > 0, \varepsilon_z < 0$) and Type II ($\varepsilon_{x,y} < 0, \varepsilon_z > 0$) HMMs. (**b**) Illustration of multilayer structure composed of metal and dielectric components. (**c**) Fishnet and (**d**) metasurface designs based on multilayered HMMs [19,21]. (**e**) Illustration of nanowire metamaterial composed of metal nanowires with dielectric matrix. (**f**) Ag nanowires in $Al_2O_3$ matrix [24]. (**g**) Metallic nanohole array [28,29]. Reproduced with permission, ACS Publications, Wiley, and The Optical Society.

## 2. Nanostructure Fabrication

Realizing as-proposed metamaterial designs for demanding nanodevice applications poses challenges to the existing fabrication techniques. Electron beam lithography (EBL), one of the most favorable methods of fabricating pre-designed metasurfaces with improved yield and resolution, is still limited by a small patterning area and relatively high cost [36,37]. An alternative method to fabricate wire metamaterial is to use an anodized alumina (AAO) template. By filling a secondary phase, such as metals or semiconductors, into the template using bottom-up deposition, this method is capable of producing nanorod arrays with a hexagonal order [38,39]. Both methods are capable of realizing sharp features or pattern size down to around 20 nm.

Recently, conventional thin film growth methods are gaining importance in fabricating nanocomposites or heterostructures that are comparable to the HMM geometries. A multilayered structure or superlattice can be easily achieved by alternating between two sources or targets (Figure 2a) using magnetron sputtering, pulsed laser deposition (PLD),

or molecular beam epitaxy (MBE) [40,41]. However, nanowire-in-matrix formation is more challenging considering multiple factors, including strain and lattice mismatch, wettability, and thermal stability between the two constituent phases. These factors are crucial to ensure well-shaped nanopillars and a sharp vertical interface without cation or inter-phase mixing. The wire-in-matrix configuration is also termed as vertically aligned nanocomposites (VANs), and has been realized in a wide range of oxide-oxide material systems, where one oxide phase is served as either nanopillars or nanodomains within another oxide matrix. These oxide-oxide VANs exhibit tunable multifunctionalities, such as magnetoresistance, superconductivity, and multiferroic properties [42–49]. The growth mechanism of VANs can be briefly explained as a self-assembled nucleation and growth. The target is typically composed of a mixture of two material components with pre-defined volume or weight ratio. During the growth, the matrix phase favors the layered growth mode (Frank–van der Merwe) while the secondary phase, with lower concentration, favors islanded (Volmer–Weber) or mixed growth mode (Stranski–Krastanov) (Figure 2b) [50]. The resulted width and height of the nanopillars are typically determined by the size of nucleation and thin film thickness.

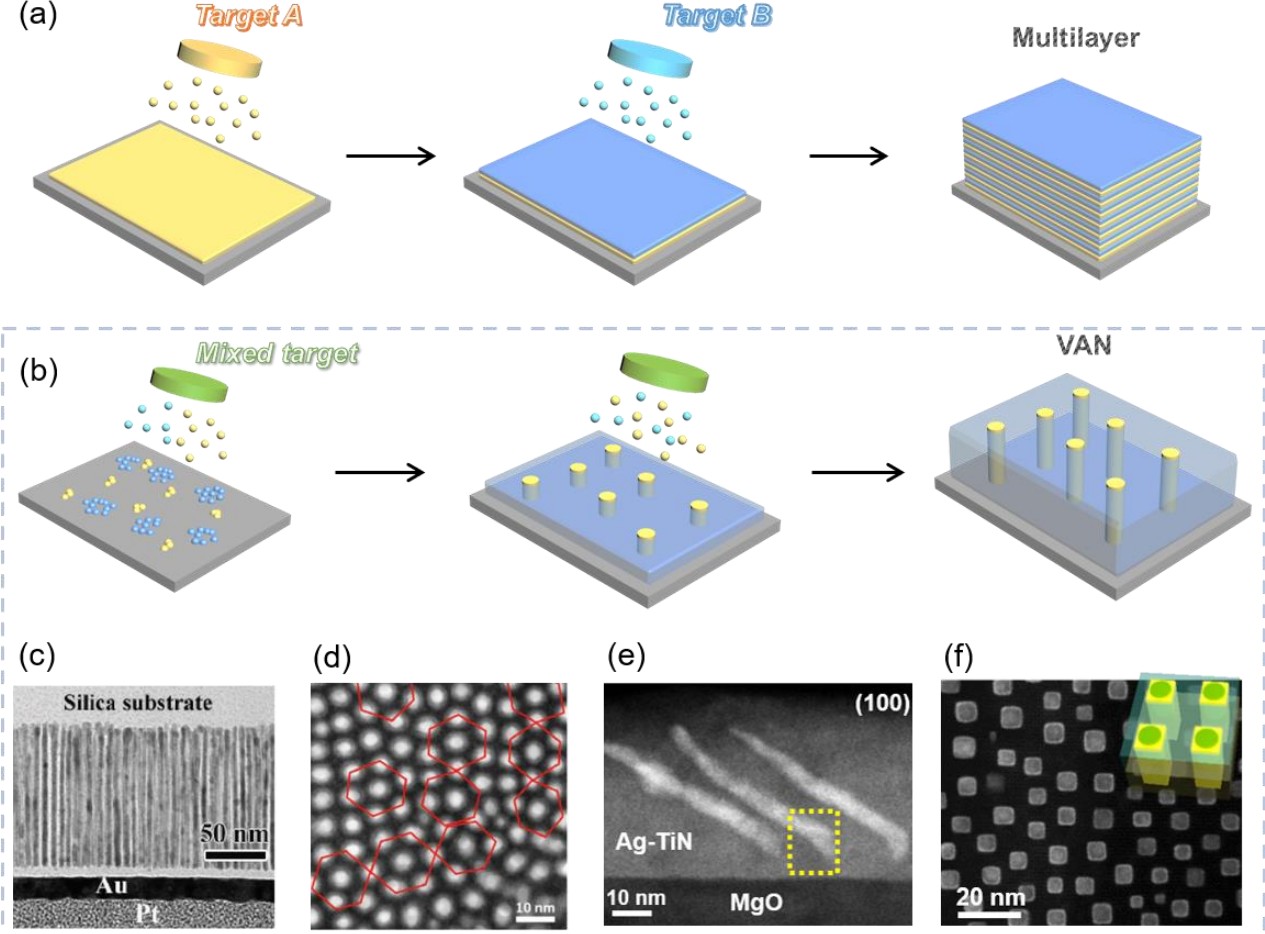

**Figure 2.** Processing of (**a**) multilayered heterostructure or metamaterial, (**b**) vertically aligned nanocomposite or metamaterial. (**c**) Ag nanowires in silica matrix [51], (**d**) Au nanopillars in TaN matrix [52], (**e**) tilted Ag nanopillars in TiN matrix [53], (**f**) highly ordered core-shell nanopillars in TiN matrix [54]. Reproduced with permission, Wiley.

Very recently, such bottom-up self-assembly has been extended to the coupling of ceramic and metals. Compare to oxide-oxide systems, involving metals brings several benefits, including (1) strong optical (e.g., Au, Ag) and magnetic (e.g., Co, Ni) properties that are drastically different from most oxides, (2) high surface energy leads to the islanded growth mode of metals as pillar-like or wire-like structures [55], (3) comparable crystal

lattice and symmetry to those of oxides, which ensure a well-coupling at the film/substrate or pillar/matrix interface. Since the first success achieved in BaTiO$_3$-Au VAN growth in 2016 [56], extensive research studies have been focused on exploring the growth mechanism, possible geometrical tunability, as well as material candidates. Examples include Ag nanowires in silica and alumina matrices (Figure 2c) [51], hexagonal-ordered Au nanopillars within TaN matrix (Figure 2d) [52], tilted Ag nanopillars in TiN matrix (Figure 2e) [53], as well as a highly ordered core-shell three-phase heterostructure (Figure 2f) [54]. All of these examples have shown great potentials in achieving various artificial metamaterial designs possessing relatively good periodicity, large-scale surface coverage, epitaxial quality, atomic-sharp interface, as well as tunable functionalities. Table 1 summarizes the recent ceramic-based wire metamaterials (or VANs) grown by physical vapor deposition technique, and their reported functionalities [51–54,56–74]. The ultimate goal is to realize an alternative method in fabricating functional heterostructures or metamaterials for applications including sensing, high-temperature plasmonics, nonlinear optics, ultrafast switching, as well as fundamental explorations, including tunable plasmonics or coupled multifunctionalities, using these highly anisotropic media.

Following the outline from Figure 3, this review covers recent advances on ceramic-based VAN metamaterials that realize tunable optical anisotropy and some additional functionalities. Selection rules for material candidates and how optical anisotropy is realized through the designs are discussed in Section 1. The atomic-scale strain coupling from the three-dimensional perspective will be covered as well to explain the formation mechanisms of such vertical aligned geometry (Section 1). Geometrical and substrate tuning and their effects on functionalities will be followed (Section 2). Additional capabilities such as sensing, magneto-optical coupling and thermal-stable plasmonics are reviewed in the Section 3. Challenges and future prospects will be discussed at the end of this review (Section 4).

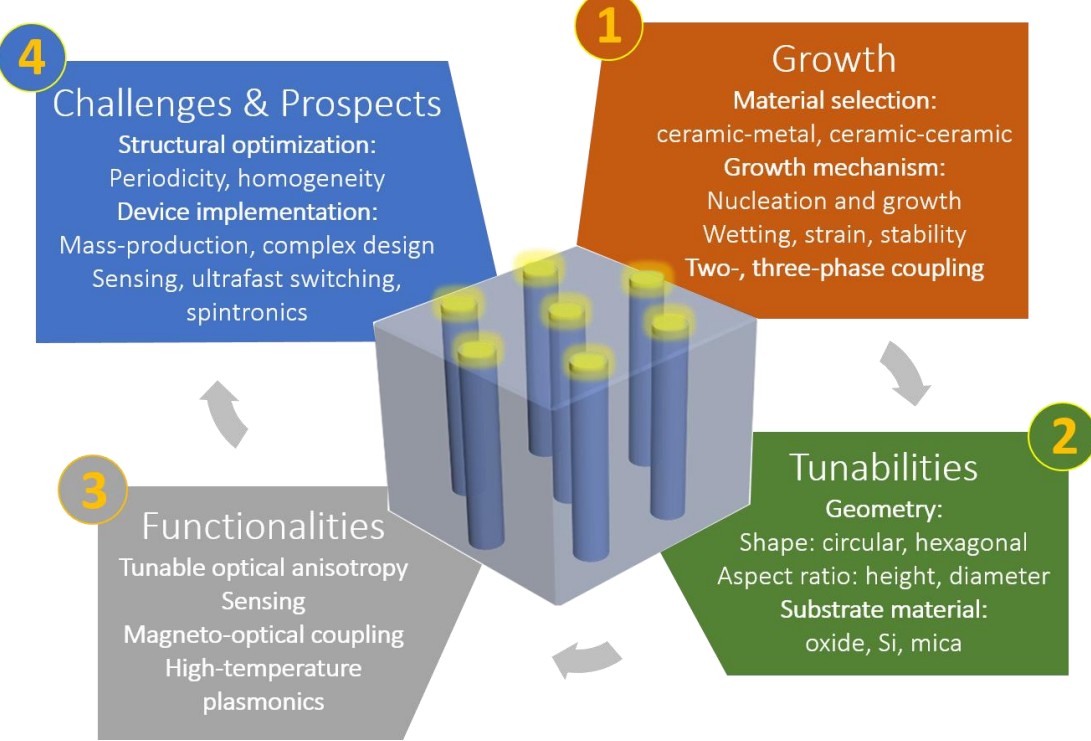

**Figure 3.** Outline. Section 1. Growth: material selection, growth mechanism, two-phase and three-phase coupling. Section 2. Tunability: geometry and substrate material. Section 3. Functionalities: tunable optical anisotropy, sensing, magneto-optical coupling, high-temperature plasmonics. Section 4. Challenges and prospects: structural optimization and device implementation.

**Table 1.** Ceramic-based VANs grown by PLD.

| Matrix Phase | Pillar Phase (2nd) | Additional Phase (3rd) | Substrate Material | Key Functionalities | Reference |
|---|---|---|---|---|---|
| $Al_2O_3$ / $SiO_2$ | Ag | - | $SiO_2$ | Plasmonic resonance, strong optical absorption | [51] |
| $BaTiO_3$ (BTO) | Au | - | $SrTiO_3$ (STO), Si (TiN/STO buffer), Mica | Tunable pillar dimension, thermal stability, hyperbolic, ferroelectric | [56–60] |
| | Au-Ag alloy | - | STO, MgO | Hyperbolic | [61] |
| | Au | ZnO | STO, MgO, $LaAlO_3$ (LAO) | Hyperbolic | [62–64] |
| ZnO | Au | - | $c-Al_2O_3$ | Tunable pillar dimension, hyperbolic | [65] |
| | Au-Ag alloy | - | $c-Al_2O_3$ | Low loss, hyperbolic | [66] |
| | Cu | - | STO, $c-Al_2O_3$ | Low loss, hyperbolic | [67] |
| $La_{0.67}Sr_{0.33}MnO_3$ (LSMO) | Au | - | STO | Tunable pillar density, | [68] |
| $La_{0.5}Sr_{0.5}FeO_3$ (LSFO) | Au | Fe | STO | Magneto-optical anisotropy | [69] |
| TaN | Au | - | MgO, STO, Si | SHG, surface-enhanced Raman scattering (SERS) effect | [52] |
| TiN | Au | - | MgO, STO, Si | Tunable pillar density, SERS effect, chemical sensing | [70,71] |
| | Ag | - | MgO, $c-Al_2O_3$ | Tunable pillar tilting, thermal stability, angular selectivity, SHG | [53] |
| | Air | - | MgO | Sensing, optical anisotropy | [72] |
| | NiO | - | MgO, Si | Tunable pillar dimension, hyperbolic, magneto-optical anisotropy | [73] |
| | NiO | Au | MgO | Magneto-optical anisotropy | [54] |

## 3. Material Selection toward Optical Anisotropy

As stated above, optical anisotropy is realized when light penetration varies along principal axes. To this end, candidates such as dielectrics and metals are considered due to their strong variation of dispersion property. Hyperbolic metamaterial certainly belongs to an extreme anisotropy scheme, where the signs of dielectric tensors along in-plane (IP, or ordinary) and out-of-plane (OP, or extra-ordinary) are completely opposite at certain wavelength region. $BaTiO_3$ ($n = 2.4$)-Au on STO substrate is the first explored ceramic-metal VAN growth. The dominating factors of this successful VAN integration lies in (1) close match between BTO, Au and STO lattices, (2) stable growth of BTO under reduced oxygen pressure (vacuum), and (3) high thermal stability and easy nucleation of Au. In terms of functionality, BTO exhibits ferroelectric property while Au exhibits strong plasmonic resonance at nanoscale [75]. As a result, the VAN structure exhibits distinct Au nanopillars and BTO matrix without intermixing or discontinuity [56,76]. The Au nanopillars possess a diameter of around 20 nm and a relatively uniform distribution as visualized from plan-view and cross-section energy-dispersive X-ray spectroscopy (EDX) mapping (Figure 4b). Effectively, the structure exhibits a Type I hyperbolic property at near infrared regime with epsilon-near-zero (ENZ) transition at 824 nm (Figure 4a). Such ENZ transition can be tuned and will be discussed later.

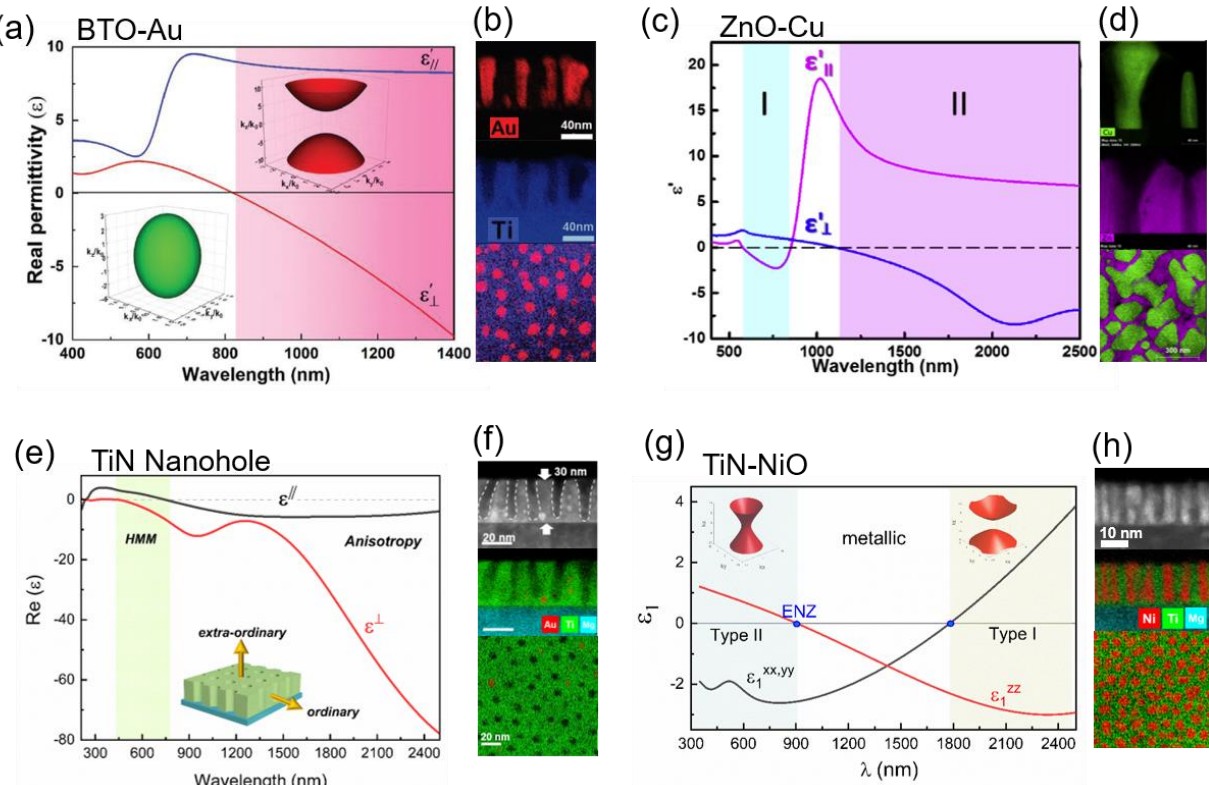

**Figure 4.** Uniaxial dielectric function and corresponding microstructures of VAN systems showing strong optical anisotropy. (**a**,**b**) BTO-Au VAN [57], (**c**,**d**) ZnO-Cu VAN [67], (**e**,**f**) TiN nanoholes [72], (**g**,**h**) TiN-NiO VAN [73]. Reproduced with permission, Wiley and ACS Publications.

The success of BTO-Au VAN paves the way for coupling a wide range of material candidates, for example, the ZnO-metal VAN system. ZnO is an interesting dielectric candidate with piezoelectric and photocatalytic properties [77]. Figure 4c,d shows a reported ZnO-Cu metamaterial where Cu nanocolumns or nanorods with dimensions around 100 nm are vertically distributed in ZnO matrix(Figure 4d) [67]. Compare to noble metals, Cu owns advantages such as cost-effectiveness and a strong localized surface plasmon (LSP) mode. Structurally, ZnO functions as a durable supporting matrix to prevent Cu from oxidation, while the ZnO/Cu interface remains atomically sharp without intermixing. Similarly, optical anisotropy with two types of hyperbolic transition are visualized at 580 to 850 nm (Type II) and above 1100 nm (Type I).

Aside from oxide-based metamaterials, transition metal nitride-based metamaterials have been proposed that also hold optical anisotropy [74]. Here, the conductive titanium nitride (TiN) matrix demonstrates several unique characteristics, including high mechanical and thermal stabilities, and plasmonic property comparable to Au. Ideally, TiN becomes a desirable supporting matrix to embed unstable nanoresonators, and potentially, TiN can be an alternative candidate to replace lossy metallic components. Using a two-step growth and chemical etching process, an epitaxial TiN nanohole film has been reported. Here, the 6-nm "air gaps" (Figure 4f) generates strong optical anisotropy as represented by the ordinary and extraordinary dielectric tensors in Figure 4e [72]. Different from conventional methods of nanohole fabrication, such as EBL or laser imprinting, using bottom-up deposition of TiN-Au VAN followed by chemical etching to remove Au nanopillars produces large-scale throughput and ultrafine holes with reasonable ordering. The chemically inert TiN provides a natural diffusion barrier against strong etching acids and results in high crystalline quality and intact film morphology.

Furthermore, the idea has been extended to coupling all-ceramic candidates to grow hyperbolic metamaterial without metals. TiN-NiO was the first explored heterostructure,

considering the natural p-type semiconducting property of NiO, as well as a close lattice match between TiN and NiO [78]. Interestingly, this heterostructure realizes a magneto-optical coupling considering the weak ferromagnetism generated in nanoscale NiO phase, which will be discussed at the end of this review. Results demonstrate an effective growth of NiO nanorods within TiN matrix (Figure 4h), with a tunable dimension by changing the pulsed laser frequency [73]. Optically, hyperbolic properties with one-fold *k*-space hyperboloid generated at (Figure 4g) below 900 nm and two-fold at above 1775 nm are confirmed. The success of growing VANs by coupling all-ceramic candidates opens tremendous opportunities since there are a wide range of functional oxides or carbides that can potentially be coupled to realize large-scale metamaterial fabrication within one-step growth.

## 4. Strain Coupling and Growth Mechanisms of VANs

Growth mechanisms of VANs, being intensively explored in oxide-oxide systems, are generally believed to be dominated by two mechanisms, i.e., nucleation and growth, and spontaneous decomposition [45,48,79]. Between the two mechanisms, the nucleation and growth mechanism covers most of the VAN self-assembly, which relies on a predefined volume ratio between the constituent phases, as well as inherit material properties such as wetting, lattice strain, as well as stability or immiscibility, which play the key role in determining the VAN growth and nucleation. Specifically, wetting or surface energy is the basis that dominates at the nucleation stage, which also shows a dependency on the crystalline plane. To realize a ceramic-metal VAN growth, the metallic phase usually favors the islanded growth mode while the oxide phase, contributing a higher volume ratio, prefers a layered growth mode [50]. In addition, lattice strain plays a role in controlling morphology and crystallinity, as in most cases, candidates with close lattice matching are considered. A third factor to consider is the stability, thermal stability is the major concern and is one crucial prerequisite for nanodevice integration. Any intermixing between the cations or any interdiffusion is not expected. These are the reasons why Au becomes the most favorable candidate as compared to metals that are easier to oxidize or melt like Aluminum.

As a result, the BTO-Au coupling is a typical example that meets the above three growth requirements. The lattice constant between Au ($a = 4.080\,A$), BTO ($a = 3.992\,A$) and STO substrate ($a = 3.905\,A$) are so close that epitaxial coupling is facilitated along the vertical BTO/Au interface as well as the lateral film/substrate interface. High resolution scanning transmission electron microscopy (HRSTEM) micrographs confirm the atomically sharp interface between Au and BTO, and a smooth transition due to the ideal lattice match (Figure 5a–e). $La_{0.67}Sr_{0.33}MnO_3$ (LSMO, $a = 3.873\,A$) is another perovskite candidate that can be coupled with Au as VAN, plus, LSMO is highly functional in terms of its magnetic and transport properties toward spintronic devices [80,81]. Results from Figure 5f,g indicate a crystalline sharp LSMO/Au interface with two types of Au domains with a 45° rotation [68]. Instead of squared-like pillar geometry for coupling between two cubic phases, the growth of ZnO and metal results in a hexagonal pillar geometry (Figure 5h,i), which is affected by the three-fold symmetry of wurtzite ZnO and the supporting sapphire (*c*-cut) substrates. The metallic phase is orientated at (111) to match the symmetry and lattice spacing of (0002) ZnO and (0006) $Al_2O_3$.

Using the two-phase VANs as a well-ordered template, more complex heterostructures such as three-phase coupling by adding a small fraction of one additional component have been explored. For example, the growth of BTO-ZnO-Au VAN (Figure 5j,k) is achieved by using the BTO-Au VAN as the template layer to grow BTO-ZnO VAN [62]. Interestingly, as shown in Figure 5j, the unique "nanoman-like" nanostructure is affected by the "vapor-liquid-solid (VLS)" growth mechanism [82,83], where Au nanorods at the bottom act as seeds for catalyzing ZnO nanowires, which are also capped on top. Similarly, the growth of a TiN-NiO-Au three-phase heterostructure is shown in Figure 5l,m [54]. By applying the highly ordered TiN-Au VAN template, the NiO pillars being nucleated on top of Au exhibit significant enhancement of long-range ordering and homogeneity. Here, a special core

(NiO)-shell (Au) nanostructure is formed via a strain compensation mechanism where the upward diffusion of Au adatoms releases the strain energy between TiN and NiO, forming the two-atomic layer shell.

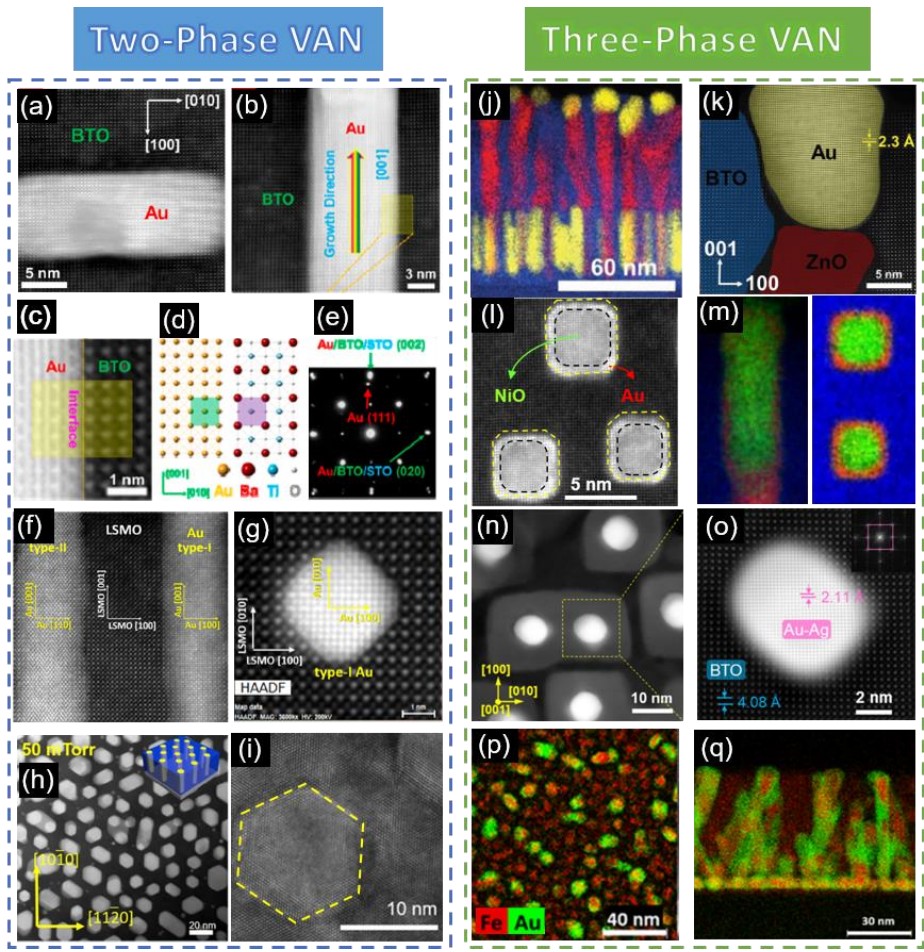

**Figure 5.** Strain coupling of two-phase and three-phase VANs. (**a**–**e**) BTO-Au VAN: (**a**) plan-view and (**b**) cross-sectional STEM micrographs at BTO/Au interface [56]. (**c**,**d**) HRSTEM micrograph at BTO/Au interface and the atomic construction. (**e**) Diffraction pattern. (**f**) Cross-sectional and (**g**) plan-view STEM images of LSMO-Au VAN [68]. (**h**) Low-magnification plan-view STEM and (**i**) plan-view HRTEM image of ZnO-Au VAN [65]. (**j**,**k**) Cross-sectional EDX mapping and HRSTEM image of BTO-ZnO-Au three-phase VAN [62]. (**l**) Plan-view HRSTEM and (**m**) corresponding EDX mapping of TiN-NiO-Au three-phase heterostructure exhibiting core-shell nanopillars [54]. (**n**,**o**) Plan-view STEM images of BTO-Au$_x$Ag$_{1-x}$ VAN [61]. (**p**) Plan-view and (**q**) cross-sectional EDX maps of LSFO-Fe-Au three-phase VAN [69]. Reproduced with permission, Wiley and ACS Publications.

Another type of complex heterostructure is simply through the mixing or alloying of two metallic components, such as Au$_x$Ag$_{1-x}$ alloy to lower the ohmic loss or to enhance the plasmonic resonance through introducing Ag [84], as well as the Au-Fe mixture to enhance the magneto-optical coupling via involving a magnetic phase [69]. The BTO-Au$_x$Ag$_{1-x}$ VAN in Figure 5n,o displays an interesting nano-domino-like structure, where the strain energy from Au$_x$Ag$_{1-x}$ alloyed nanopillars changes the lattice spacing of the surrounding BTO matrix, forming extruded and ordered nanodomains. On the other hand, the La$_{0.5}$Sr$_{0.5}$FeO$_3$-Fe-Au nanocomposite as shown in Figure 5p,q demonstrates a successful coupling between Fe and Au as relatively well-aligned nanopillars without intermixing, which can be traced to the phase stability and relative weight fraction at the growth temperature.

## 5. Tuning Geometry and Substrate

Besides optical anisotropy, a wide range of tunabilities has been investigated through the control over geometry or substrate of the ceramic-based VANs. Geometrical control for tunable plasmonics has been extensively reported in chemically grown nanostructures [85,86]. Here, by tailoring the growth parameters such as laser frequency, duration of growth, second phase concentration, temperature, oxygen pressure, subsequent tuning of aspect ratio, shape, distribution, and morphology can be effectively realized using PLD [87]. A thickness dependent BTO-Au growth has been reported by Zhang et al [57]. Simply by changing the duration of growth, the aspect ratio of Au nanopillars are tuned from 3.6 to 0.8, with a morphological change from nanopillar to nanodisk array (Figure 6a). The aspect ratio of Au is primarily affected by the height of the pillars. As a result, a continuous blue shift of the hyperbolic transition (epsilon-near-zero, ENZ) of the extraordinary dielectric tensors (Figure 6b,c) is observed, which indicates a change of charge carrier density of the entire heterostructure. Controlling laser frequency for tuning nanocolumn dimensions has been studied in many oxide-oxide VANs [43]. An example shown in Figure 6e–g is the tuning of ZnO-Au VAN. Here, reducing the laser frequency lowers the growth rate and allows for a longer resting time of Au adatoms. The resulting nanopillars are much wider and thicker (Figure 5d,f). Optical penetration is extremely sensitive to the change of nanoresonators, therefore, effective changes of inter-pillar distance and size both act on tuning of dielectric function (Figure 6e,g). There are certainly other tuning configurations, such as the TiN-Ag metamaterial as shown in Figure 2e [53]. In this study, the angle of Ag nanopillar is effectively changed from 0° to 50° by controlling the growth rate. The tilted resonators realize a strong angular selectivity and nonlinear response over a wide spectrum range, excited by the strong Ag LSP mode with respect to the angle of incident light.

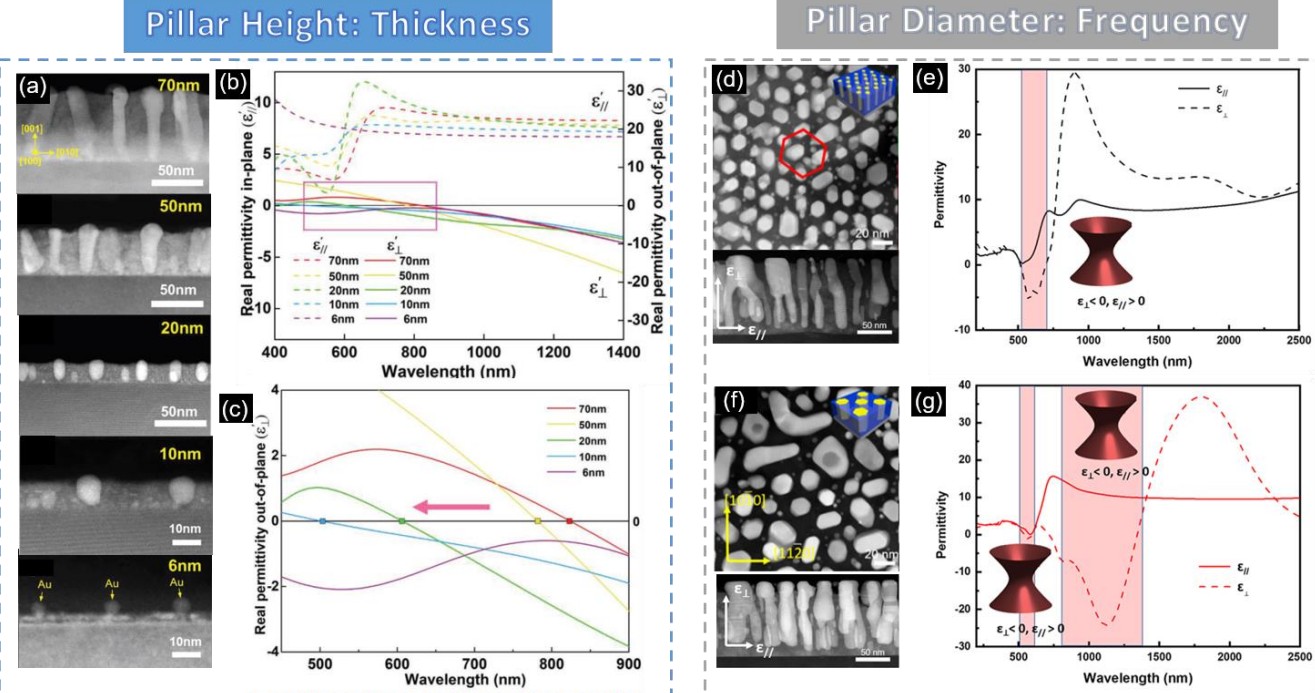

**Figure 6.** Geometrical tunabilities. (**a**) BTO-Au with change of film thickness, and effectively, Au phase is tuned from nanopillars to nanodisks. (**b,c**) Effective tuning of hyperbolic property, OP dielectric tensor exhibits continuous blue shift with reduction of film thickness [57]. (**d,f**) STEM micrographs of ZnO-Au VAN with tuning of pillar diameter and (**e,g**) effective changes of uniaxial dielectric tensors [65]. Reproduced with permission, Wiley and ACS Publications.

Since bottom-up deposition realizes inch-scale surface coverage and high crystallinity as compared to the existing top-down approaches such as nanolithography, extending the growth on Si or flexible substrates is extremely crucial toward on-chip and flexible opto-electronic devices [88]. To this end, the growth of BTO-Au VAN has been explored on buffered Si substrate [59]. Here, the TiN/STO (<20 nm) buffer layer serves as a transition layer to establish the thin film epitaxy on Si and to enhance the overall growth morphology (Figure 7a). TiN buffered Si integration has been reported in many oxide-oxide systems such as LSMO-NiO [89,90]. Direct nucleation of two-phase nanocomposite would be rather challenging since the surface energy, crystal symmetry, and lattice parameter of Si could be different from oxides and metals. From Figure 7b–f, the coupled X-ray diffraction (XRD) and TEM results indicate well-distributed Au nanopillars within BTO matrix, while the BTO/STO/TiN/Si lateral interfaces remain crystalline sharp without interdiffusion.

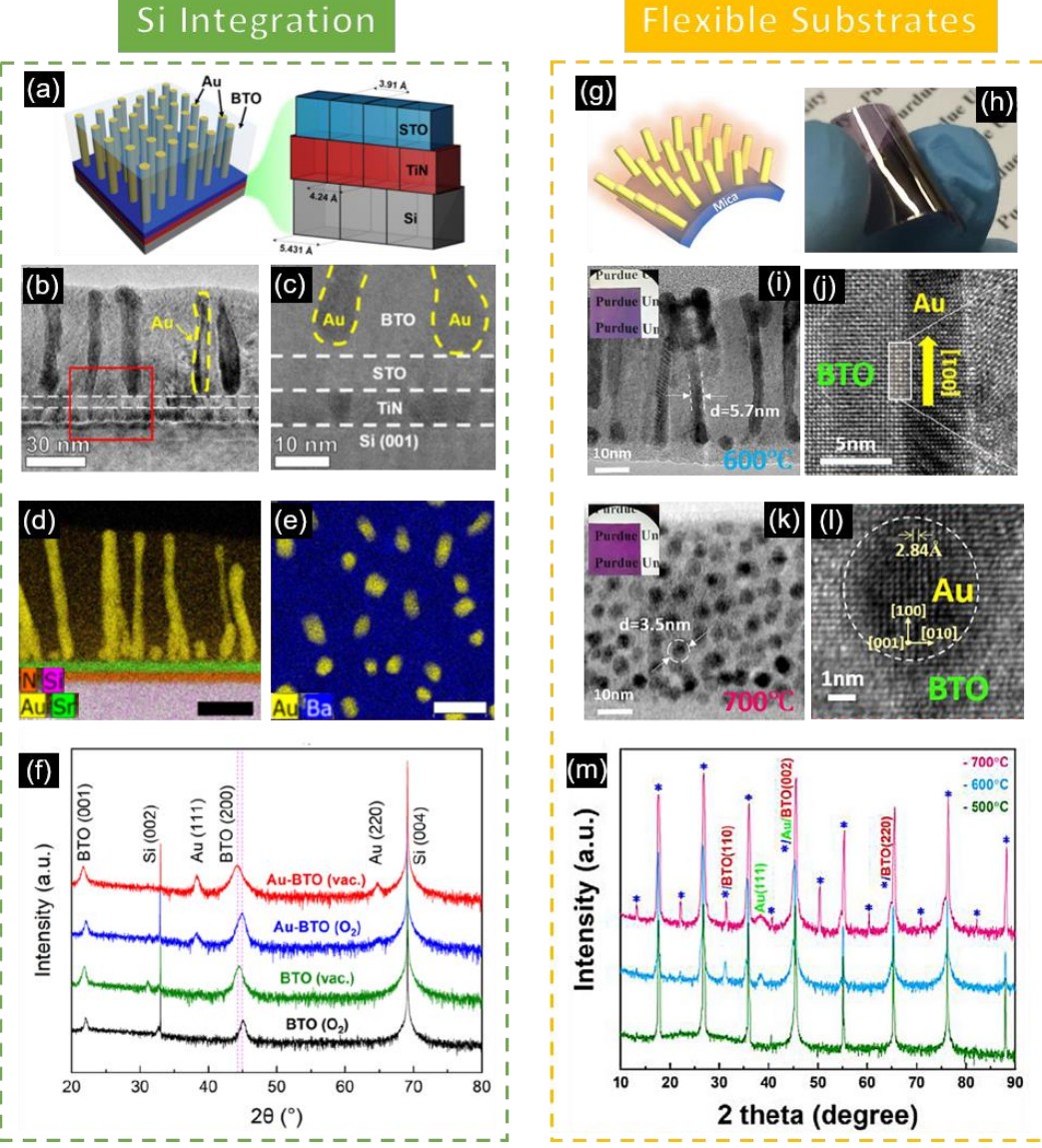

**Figure 7.** (**a–f**) BTO-Au VAN integrated on buffered Si substrate [59]. (**a**) Buffer layer is composed of STO/TiN bilayer. (**b**,**c**) Cross-sectional TEM images of overall microstructure and at the lateral interface region, (**d**) corresponding cross-sectional and (**e**) plan-view EDX maps. (**f**) XRD θ-2θ scans showing high epitaxial quality. (**g–m**) BTO-Au VAN integrated on flexible mica substrate [60]. (**h**) Light reflection image upon bending the sample. (**i–j**) TEM images of Au nanopillars in BTO matrix showing epitaxial strain coupling. (**k**,**l**) TEM images of Au nanoparticle in BTO matrix under higher growth temperature. (**m**) XRD θ-2θ scans exhibiting high crystalline quality of the films. Reproduced with permission, ACS Publications and Elsevier.

Flexible muscovite mica, with inter-plane van der Waals bonding, serves as an ideal template to grow functional thin films for wearable and flexible devices [91–93]. Recently, Liu et al. demonstrated the successful integration of BTO-Au VAN directly on mica substrate, with epitaxial growth and sharp atomic interface [60]. At the macroscopic scale, the entire sample is bendable without fracture or failure (Figure 7a,b). Interestingly, the change of growth temperature reflects a morphological transition from nanopillars to nanoparticles, which is potentially owing to the high sensitivity of the nucleation energy of BTO and Au with respect to the mica substrate. A relatively stable optical performance is achieved after cycles of concave and convex bending, which is promising for further investigations on material integration and device designs.

## 6. Beyond Optical Anisotropy

The ultimate goals of ceramic-based metamaterial using conventional bottom-up growth are certainly not limited to tunabilities and optical anisotropies. Here, we present few examples on coupling growth designs to realize capabilities such as sensing, magneto-optical coupling, as well as high-temperature plasmonics. As briefly mentioned in the earlier discussion, the design of plasmonic TiN nanoholes is realized through removal of Au phase from a highly-ordered TiN-Au VAN template (Figure 8a) [72]. The nanoscale air holes generate strong LSPs close to the edge of TiN metasurfaces, which are highly sensitive to the change of local refractive indices and can be applied for sensing. The first demonstration of sensing was conducted by coupling the surface with 2D perovskite nanoplates to resolve real-time changes of photoluminescence (PL). Compare to pure TiN film and TiN-Au template, the PL signal from TiN nanoholes with 2D nanoplates exhibits an obvious enhancement, while three new peaks located at 432, 456, and 514 nm indicate additional recombination states affected by coupling with such defective topology (Figure 8b,c) [94]. For the second demonstration, the transmittance measurement was conducted by dropping the immersion oil with varied refractive index onto the sample surface. An obvious redshift (638.56 to 742.21 nm) upon the increase of refractive index (1.35 to 1.75) is observed (Figure 8d), indicating a change of surface plasmon (SP) mode corresponding to the local change of refraction index. The calculated figure of merit (FOM) and refractive index unit (RIU) can be enhanced by improving the etching efficiency or by enlarging the size of the nanoholes.

Magneto-optical coupling, on the other hand, plays an important part in realizing ultrafast switching, spintronic devices, and all-optical photonic circuits [95,96]. An intuitive method is to couple magnetic and optical components at nanoscale, in which sense the VAN provides an ideal and promising way to realize such designs. Design of TiN-NiO-Au three-phase heterostructure is presented in Figure 8e,f, it shows a magnetic core–plasmonic shell coupling with a high degree of periodicity [54]. The growth mechanism has been explained in the previous section, and as a result, such coupling realizes a strong enhancement of Kerr signal, which also exhibits certain anisotropy owing to the vertically coupled interface. The results indicate a weak ferromagnetic behavior of the NiO when its dimension reduces to few nanometers. It is noted that the shape of P-MOKE hysteresis is rather irregular owing to the two contributing coupling schemes (Figure 8g), i.e., the coupling between weak ferromagnetic NiO core and plasmonics Au shell, as well as the ferromagnetic Ni nanodomains at the bottom layer.

Another characteristic, which is also crucial in the field of plasmonics, is to design durable nanostructures for solar-cells, biomedical sensors, and quantum computing technologies [34,97–99]. From a materials perspective, metals such as Ag and Cu exhibit strong surface-enhanced Raman scattering and SP modes, while their poor thermal stability and chemical reactions potentially deteriorate the device performance under environmental fluctuations. By embedding Ag nanopillars in a durable TiN matrix, a strong enhancement of mechanical and thermal stability has been realized. Figure 8h shows the STEM and diffraction patterns (DP) of the nanostructure after a heating cycle of >500 °C [53]. The TiN/Ag interface remains intact, and the sharp DPs indicate the high crystallinity.

Interestingly, a tilted Ag nanopillar growth is realized, which results in several interesting features, such as second harmonic generation (SHG) and selective reflectance from UV to mid-infrared regime (Figure 8i), as mentioned earlier.

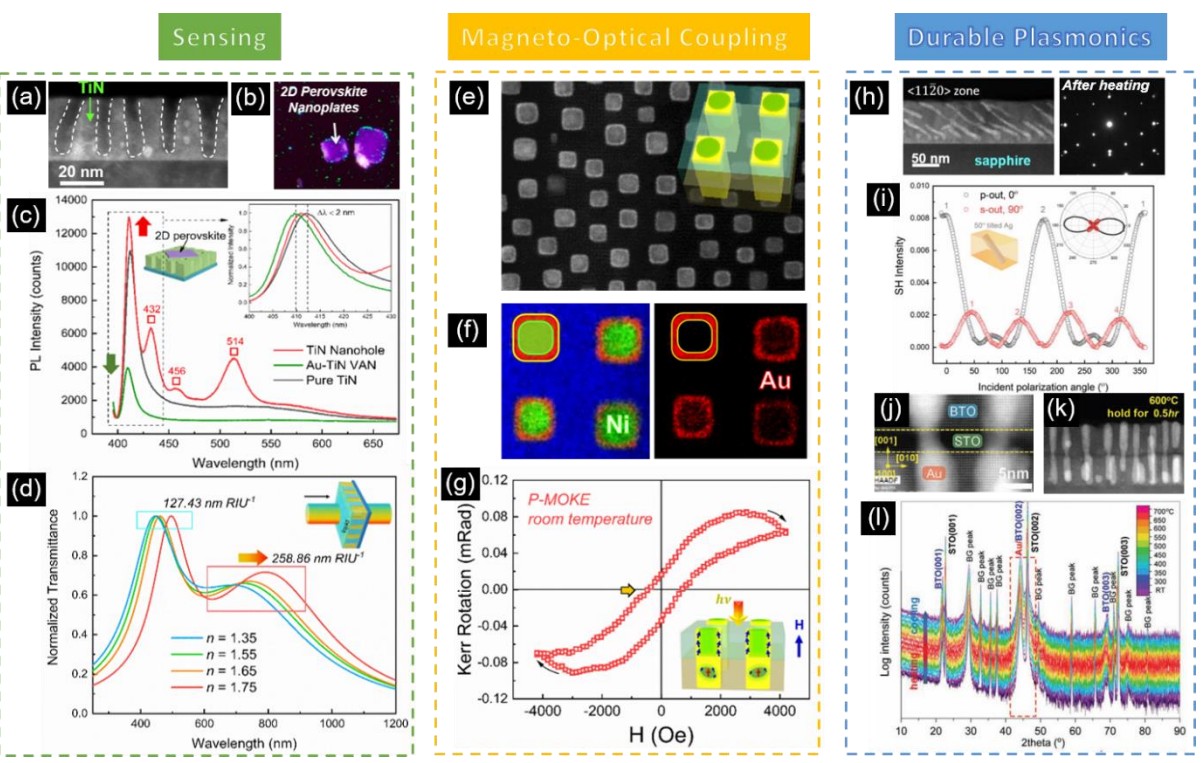

**Figure 8.** Multifunctionalities. (**a**) Cross-sectional STEM micrograph of TiN nanohole film [72]. (**b**) Crystallization of 2D perovskite nanoplates on TiN nanohole surface. (**c**) PL spectra comparing 2D nanoflakes grown on TiN nanohole, TiN-Au VAN and pure TiN film. (**d**) Refractive index sensing of TiN nanohole film. (**e**) Plan-view STEM micrograph of TiN-NiO-Au three-phase VAN showing highly ordered core-shell nanopillars [54]. (**f**) EDX mapping of Ni (green), Au (red) and Ti (blue). (**g**) P-MOKE hysteresis loop. (**h**) STEM morphology and DPs of TiN-Ag metamaterial after heating of above 500 °C [53]. (**i**) SHG pattern of TiN-Ag. (**j**) HRSTEM image of STO interlayered BTO-Au sandwich [76]. (**k**) STEM micrograph upon in-situ heating at 600 °C. (**l**) Ex-situ XRD patterns with changes of temperature. Reproduced with permission, ACS Publications and Wiley.

Another consideration is the thermal stability at ceramic/metal interface. There are few studies investigating the thermal stability of some oxide-metal VANs, which have provided some promising feedback. For example, a complex interlayered BTO-Au/STO/BTO-Au heterostructure is displayed in Figure 8j [76]. In-situ thermal stability test is conducted inside a TEM column where no obvious morphological change is observed upon 600 °C annealing. Additional temperature dependent ex-situ XRD characterizations indicate highly stable diffraction peaks without obvious change of position or full-width half maximum (FWHM) values. Though the demonstrated high thermal stability may not be applicable to some systems involving unstable metals (e.g., Al, Cu), these studies on oxide-Au present clear evidence that the Au contained VAN structures are suitable for high-temperature plasmonics.

## 7. Challenges and Opportunities

As an alternative fabrication of top-down lithographic patterning, the bottom-up growth by self-assembling of two or more functional phases realizes comparable geometry as wire metamaterials, meanwhile possessing a wide range of tunabilities in terms of geometry, materials, and functionalities. Besides the common dielectric-metal designs, novel heterostructures, such as TiN nanohole, all-ceramic VAN (TiN-NiO), three-phase

coupling (BTO-ZnO-Au), and highly ordered core-shell nanopillars, can realize extreme optical anisotropy, magneto-optical coupling, as well as microfluidic sensing. These designs pave the way for advanced artificial metamaterials in overcoming challenges such as thermal instability, inhomogeneity, ohmic losses. Originating from thin film sciences, the growth of functional nanocomposites using conventional method presents fascinating prospects for next-generation nanophotonics and spintronics. Meanwhile, there are still remaining challenges to be resolved.

Structural optimization: Aside from multiple interesting and novel coupling schemes that present outstanding flexibility and additional advantages over top-down approaches, most of the reported structures are still at an early stage. Thus, there are several questions or concerns that demand additional exploration, both experimentally and theoretically. How is the reproducibility of the VAN growth? How can we improve the uniformity and periodicity of the nanopillars? Is there any intermixing at the vertical interface? Indeed, improving uniformity is one of the most urgent tasks. From the cross-section STEM micrographs, there are always observable nanopillars nucleated at the initial growth stage, but stopped in the middle of the growth, resulting in dense nanoparticles at the bottom section of the film. From the top-view, the distribution of the nanopillars is not as ideal as the EBL patterned features, as there is always inhomogeneity related to either shape or diameter. These factors become more serious when extending the growth to other substrates or involving multiple phases, when the strain and surface energy add extra perturbations to the growth. Therefore, optimizing the quality of the VAN structure by carefully exploring the growth parameters and related material properties, adding additional processing to facilitate ordering, and coupling with theoretical models to predict the strain and growth condition are some valuable future directions. Improving the periodicity using a highly ordered template, substrate treatment, and defect assisted nucleation have been demonstrated as effective methods [54,100,101].

Toward nanodevices: How to effectively implement such novel heterostructures into practical devices or applications is another challenge to address. Effective tuning of physical properties, such as plasmonic resonance, hyperbolic transition, magneto-optical coupling, magnetic hysteresis, and thermal stability, have been demonstrated, however, these studies are limited to fundamental research. Compared to continuous pure films, involving more interfaces could potentially bring defects, strain, or instability that could cause light scattering, current leakage or tunneling, and intermixing, especially upon environmental fluctuations. In parallel of investigating those potential issues, one can take advantage of such as-grown metamaterials to realize nanoscale coupling of multifunctional phases with inch-scale coverage, to avoid redundant fabrication processes and achieve mass-production of nanodevices. So far, some successful demonstrations of sensing have been achieved, including TiN-Au metasurface for chemical bonding detection [70], microfluidic sensing using TiN nanoholes [72], and TiN-Ag nanocomposite as a high-temperature angular reflector [53]. These studies present great potential and pave the way for applying similar tests to other oxide-metal VANs systems.

The current fabrication of nanodevices rely heavily on lithography techniques. To implement these ceramic-metal VANs for future applications such as spintronics, tunneling junctions, magneto-optical switching, or high-temperature sensors, there are two major advantages to consider. First, bottom-up self-assembly allows the coupling of a wide range of material candidates within metal and ceramic families. If the growth parameters for a specific VAN configuration (e.g., density, aspect ratio) can be traced according to extensive experimental results using a computing or modeling method, many possible VAN systems (A + B) can be easily designed and fabricated. Such self-assembled nanostructures could minimize the complexity or cost of fabrication. Toward complex nanodevice designs such as the already demonstrated double-fishnet hyperbolic metamaterial or patterned tunnel junctions [18,102], these natural two-phase or multiphase geometries, including VAN, nanoparticle-in-matrix, or superlattice structure, can be ideal templates to be coupled with nanolithography methods toward plasmonics metamaterial designs.

**Author Contributions:** Conceptualization, X.W. and H.W.; writing—original draft preparation, X.W.; writing—review and editing, X.W. and H.W.; funding acquisition, H.W. All authors have read and agreed to the published version of the manuscript.

**Funding:** The review covers a wide range of metal-oxide nanocomposite systems that were supported the U.S. Department of Energy, Office of Science, Basic Energy Sciences under Award DE-SC0020077. X.W. and H.W. acknowledge the funding support from U.S. National Science Foundation, DMR-2016453 for nitride VANs and DMR-1565822 for oxide-oxide VANs.

**Institutional Review Board Statement:** Not applicable.

**Informed Consent Statement:** Not applicable.

**Conflicts of Interest:** The authors declare no conflict of interest.

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
