# Peer review of "Recent Advances in Vertically Aligned Nanocomposites with Tunable Optical Anisotropy: Fundamentals and Beyond"

_chemosensors, doi:10.3390/chemosensors9060145_

Round 1

Reviewer 1 Report

This is well-prepared nice review manuscript. I recommend publication of this manuscript in Chemosensors with some general revisions. Please see below.

1) Although this manuscript provides nice figures for each topic, figures for basic and general guidance are not sell included. It would be a good idea to add one initial figure to show outline of scientific contents in this review article.

2) One figure with own opinions by the authors can be added to Conclusion section.

3) Relations with sensors have to be more emphasized in both Introduction and Conclusion. Recent comprehensive papers on usages of metamaterials and related structures in sensor application had better be added more (for example, see, https://www.journal.csj.jp/doi/abs/10.1246/bcsj.20180236, https://pubs.rsc.org/en/content/articlelanding/2020/NR/C9NR08433A#!divAbstract)

Reviewer 2 Report

Authors have written a good manuscript describing recent advancements in the field of vertically aligned nanocomposites. Manuscript is well written and describes good range of studies. However, few things authors should add further in the manuscript.

There is a need to elaborate a little bit further on the underlying mechanism with examples and support from the literature. Right now, it is described briefly without any support for the proposed mechanism (page 7).

Further, on page 6, the text is a bit confusing due to the wrong reference to figures. The first paragraph refers to Figure 3(e), 3(f) and second paragraph refers to figure 3(g) and 3(h) which do not exist.

Authors should describe the existing examples of the applications briefly in the main text as well. Right now it is just briefly mentioned in the conclusion section.

Authors should further describe the recommendations for the further development of the field (with reasons and their opinions) in the conclusion.  For example, from material, technology and/or application perspective. 

Reviewer 3 Report

The work presented is well structured and very interesting.

Author Response

We thank for the reviewer's comments. All the revisions are highlighted (red color) in the revised manuscript for your reference. 

Reviewer 4 Report

The review paper entitled "Recent advances in vertically aligned nanocomposites with tun-2 able optical anisotropy: fundamentals and beyond" by Wand and Wang, reports on a great variety of nanostructures with vertical growth that combine different dielectric and plasmonic materials. 

It includes a thorough list of different materials and techniques and of their potential applications. The review focuses mainly on the optical anisotropy of the nanostructures but also covers other properties such as magneto-optical and plasmonic properties, which are crucial for their applicability. Also the influence of substrate selection and the possibilities of tuning the optical anisotropy are discussed.

The review is well written and structured. The list of references can be very useful to the readers of the Chemosensors and of even a broader audience.

I think the paper can be published as is, although minor changes could improve it:

- In figure 2 caption, parc (c), Ag nanowires in alumina matrix are described. Even though the cited reference (49) deals with such a structure, I think the picture corresponds to a different structure, also discussed in the reference.

- In lines 114-115, the expressions 'and to correlate effective changes of functionalities' is confusin to me. Can be clarified?

- In line 120 there is a typo: 'principle' is written instead of the correct 'principal'

Yours sincerely
